# Effects of different emission inventories on tropospheric ozone and methane lifetime

Catherine Acquah<sup>1</sup>, Laura Stecher<sup>1,a</sup>, Mariano Mertens<sup>1,2</sup>, and Patrick Jöckel<sup>1</sup>

<sup>a</sup>now at: Yusuf Hamied Department of Chemistry, University of Cambridge, Cambridge, CB2 1EW, United Kingdom

Correspondence: Patrick Jöckel (Patrick.Joeckel@dlr.de)

Abstract. This study assesses the influence of anthropogenic emission inventories of ozone  $(O_3)$  precursor species (i.e.,  $NO_x$ , CO and NMHCs) prescribed in the simulations of the two phases of the Chemistry-Climate Model Initiative (CCMI) on tropospheric  $O_3$ , the hydroxyl radical (OH) and the methane (CH<sub>4</sub>) lifetime. We performed two transient simulations for the period 2000 - 2010 with the chemistry-climate model EMAC, one prescribing the emission inventory of CCMI-1, and the other that of CCMI-2022. Using the tagging approach, we attribute the differences of  $O_3$ , OH and the tropospheric  $CH_4$  lifetime to individual emission sectors. It is, to our knowledge, the first application of the tagging approach to attribute changes of the simulated  $CH_4$  lifetime to individual emission sectors.

The emission inventory used for CCMI-2022 leads to a 3.7% larger tropospheric  $O_3$  column, and to a 3.2% shorter tropospheric  $CH_4$  lifetime compared to CCMI-1 in the northern hemisphere. In the southern hemisphere, the tropospheric  $O_3$  column is 4.5% larger, and the tropospheric  $CH_4$  lifetime 4.3% shorter. Differences in tropospheric  $O_3$  are largely driven by changes of emissions from the anthropogenic non-traffic and land transport sectors in the northern hemisphere. In the southern hemisphere, the primary contributors are emissions from anthropogenic non-traffic, biomass burning and shipping. These sectors also play a significant role in reducing the simulated tropospheric  $CH_4$  lifetime. However, the contribution of a particular sector to changes in  $O_3$  does not necessarily align with its impact on the  $CH_4$  lifetime.

Copyright statement. TEXT

#### 1 Introduction

Tropospheric ozone  $(O_3)$ , a greenhouse gas, plays a significant role in influencing both, climate and air quality (Haagensmit, 1952; Stevenson et al., 2013). It adversely affects human health and vegetation. Anthropogenic activities, including industry and land transport, release  $O_3$  precursors such as nitrogen oxides  $(NO_x)$ , non-methane hydrocarbons (NMHCs), and carbon monoxide (CO), affecting air quality. Model data indicate an increase of about 30% of the total mass of tropospheric  $O_3$  between 1850 and 2010 (Lamarque et al., 2005; Young et al., 2013). As a greenhouse gas in the middle and upper troposphere,

<sup>&</sup>lt;sup>1</sup>Deutsches Zentrum für Luft- und Raumfahrt, Institut für Physik der Atmosphäre, Oberpfaffenhofen, Germany

<sup>&</sup>lt;sup>2</sup>Faculty of Aerospace Engineering, Section Operations and Environment, Delft University of Technology, 2629 HS, Delft, The Netherlands

O<sub>3</sub> contributes to positive radiative forcing and consequently to global warming (Stevenson et al., 2013). Next to its role as a greenhouse gas, tropospheric O<sub>3</sub> is one key component of photochemical smog (Haagensmit, 1952). As an air pollutant, during prolonged periods of fair-weather conditions, high ground level O<sub>3</sub> concentrations can have adverse effects on human health (Nuvolone et al., 2018). Some studies have already demonstrated an increased mortality after prolonged exposure (Bell et al., 2004; Jerrett et al., 2009; Silva et al., 2013). Concentrations of tropospheric O<sub>3</sub> and its precursors are significantly elevated in many regions, not only due to road traffic (Niemeier et al., 2006), but also due to industrial processes, biomass burning as well as biogenic emissions (Seinfeld and Pandis, 2016). Another relevant aspect relates to the phytotoxicity of tropospheric O<sub>3</sub>, as it is one of the most potent air pollutants for plants (Ashmore, 2005; Wedow et al., 2021). Various studies have demonstrated the reduction of agricultural crop yields (Emberson et al., 2018; Iglesias et al., 2006) and damage to broadleaves (Moura et al., 2014). The photochemical production of  $O_3$  in the troposphere is non-linearly dependent on various precursors, such as  $NO_x$ , CO, and NMHCs (Bourtsoukidis et al., 2019; Crutzen, 1974). NO<sub>x</sub> emissions from anthropogenic sources originate e.g. from industry, land transport, aviation, and shipping (e.g., Uherek et al., 2010). Natural NO<sub>x</sub> is emitted from soil and produced by lightning discharges (e.g., Schumann and Huntrieser, 2007; Vinken et al., 2014). According to Seinfeld and Pandis (2016), two-thirds of CO is emitted from anthropogenic activities, including the oxidation of anthropogenically produced methane (CH<sub>4</sub>). Natural emissions of CO occur during the combustion of biomass, e.g. lightning induced fires (Zheng et al., 2019), and from the oxidation of hydrocarbons, but also from plants and the oceans (Khalil and Rasmussen, 1990). The origin of natural NMHC emissions are marine and terrestrial environments, e.g., vegetation (Guenther et al., 1995). The main anthropogenic sources of NMHC include the incomplete combustion of fossil fuels, petroleum from geological reservoirs, and the distillation and distribution of oil and gas products (Pozzer et al., 2010).

An important aspect of tropospheric  $O_3$  is its role as a source of the hydroxyl radical (OH), which is a very relevant oxidant in the atmosphere (Monks, 2005), and thus largely determines the self-cleansing capacity of the troposphere (Monks et al., 2015). The abundance of OH influences the atmospheric lifetime of the greenhouse gas CH<sub>4</sub>, as the oxidation with OH represents the most important sink of atmospheric CH<sub>4</sub> (Lelieveld et al., 2016; Voulgarakis et al., 2013; Saunois et al., 2020). CH<sub>4</sub> decomposition by OH is considered as a source of NMHCs (Seinfeld and Pandis, 2016), which in turn are part of tropospheric O<sub>3</sub> production process (Crutzen, 1974), and thus the OH formation links the processes of CH<sub>4</sub> decomposition and tropospheric O<sub>3</sub> production and vice versa. According to the IPCC Sixth Assessment Report (IPCC, 2021) CH<sub>4</sub> is the second-most important greenhouse gas directly emitted by human activity. Different studies demonstrate the wide range of uncertainty of CH<sub>4</sub> lifetime estimates, which is the reason why the investigation of the CH<sub>4</sub> lifetime remains an important research topic. In particular, neither OH (on the global scale), nor the CH<sub>4</sub> lifetime, can be observed directly (e.g. Duncan et al., 2024). Therefore, estimates of the CH<sub>4</sub> lifetime are derived from observations of species with known emissions, whose main sink is the oxidation with OH, such as methylchloroform (CH<sub>3</sub>CCl<sub>3</sub>) (Krol et al., 1998; Prinn et al., 2005; Montzka et al., 2011) or <sup>14</sup>CO (Brenninkmeijer et al., 1992; Jöckel et al., 2002; Manning et al., 2005). Estimates of the CH<sub>4</sub> lifetime based on observations of  $CH_3CCl_3$  are, for example, 10.2 (+0.9 -0.7) years (Prinn et al., 2005) and 9.1  $\pm$  0.9 years (Prather et al., 2012). The CMIP6 AerChemMIP model ensemble as analyzed by Stevenson et al. (2020) suggests a whole-atmosphere CH<sub>4</sub> lifetime of  $8.4 \pm 0.3$  years for the present-day, while the two most recent IPCC Reports state a lifetime of  $9.25 \pm 0.6$  years (Myhre et al., 2013) and  $9.1 \pm 0.9$  years (Szopa et al., 2021), respectively. Anthropogenic emissions of  $NO_x$  and CO impacting the OH abundance have influenced the development of atmospheric  $CH_4$  concentrations in recent decades (Stevenson et al., 2020; Skeie et al., 2023).





To summarize,  $O_3$  precursors strongly influence tropospheric  $O_3$  and  $CH_4$  lifetime. The determination of  $O_3$  precursor emissions is not trivial, and emission inventories are subject to large uncertainties. Hoesly et al. (2018) suggest, that the uncertainties can range from  $\pm 8\%$  -  $\pm 176\%$ . The tropospheric  $O_3$ , OH and  $CH_4$  lifetime simulated by chemistry-climate models (CCMs) largely depends on the choice of the emission inventory. The two phases of the Chemistry-Climate Model Initiative (CCMI) use different emission inventories (see CCMI-1 simulation protocol; CCMI-2022 simulation protocol). CCMI-1 uses the MACCity emissions inventory (Granier et al., 2011), whereas CCMI-2022 uses the Community Emissions Data System (CEDS; Hoesly et al., 2018), which has been also used by AerChemMIP, which is endorsed in the Coupled Model Intercomparison Project 6 (CMIP6) (Collins et al., 2017). The latter multi-model comparison initiatives form an important groundwork for the IPCC reports (IPCC, 2013, 2021). Comparing the EMAC (ECHAM5/ MESSy Atmospheric Chemistry) simulations performed for the two different phases of CCMI (see Fig. 1). One reason for the shorter lifetime is the changed emission inventory of  $O_3$  precursor species.

In this study, we therefore specifically examine the effect of the modified emission inventories on the tropospheric  $O_3$  abundance and the  $CH_4$  lifetime. For this purpose, we performed targeted simulations with the EMAC model, prescribing either the emission inventory of CCMI-1 or CCMI-2022 for  $O_3$  precursor species. The tropospheric  $CH_4$  lifetime corresponding to these simulations are shown by the red curves in Fig. 1. The changed prescribed emission inventories can not explain the full difference of tropospheric  $CH_4$  lifetime between the EMAC simulations originally performed for CCMI-1 and CCMI-2022 (blue and black curves in Fig. 1). There have been further model developments and setup changes from CCMI-1 to CCMI-2022 that have, however, all been used for the targeted simulations performed for this study (red curves) and that we address in the supplement.

Further, we use the tagging approach to attribute the  $O_3$  and  $CH_4$  lifetime differences caused by the changed prescribed emissions to individual emission sectors. Some previous studies have successfully applied the tagging approach for determining the contributions of specific source categories to the tropospheric  $O_3$  (Butler et al., 2020; Grewe et al., 2012, 2010, 2017; Mertens et al., 2018, 2020; Rieger et al., 2018; Kilian et al., 2024), while the tagging method to attribute  $CH_4$  lifetime differences has, to our knowledge, been applied only once (Mertens et al., 2024).

The paper is structured as follows: Section 2 describes the used CCM EMAC shortly and explains the simulation set-up including information about the used emission inventories, the tagging method and the method to calculate the tropospheric  $CH_4$  lifetime. Section 3 shows the results, i.e. the impact of the modified emission inventory on tropospheric  $O_3$ , OH and the  $CH_4$  lifetime. It is followed by a discussion in Sect. 4 including limitations of the applied tagging method and the simulation set-up, as well as a comparison to other studies. We close with a summary of our findings and some concluding remarks in Sect. 5.

**Figure 1.** Tropospheric CH<sub>4</sub> lifetime with respect to the oxidation with OH,  $\tau_{CH_4,OH_{total}}$ , for the EMAC simulations performed for phase 1 of CCMI (CCMI-1) in blue, phase 2 of CCMI (CCMI-2022) in black, and of the simulations analyzed in the present study, EMIS-01 and EMIS-02, in red. See Jöckel et al. (2016) for details of the setup of the CCMI-1 simulations and Jöckel (2023) for REFD1.

# 2 Methods and Data


To determine the impact of the two emission inventories on tropospheric  $O_3$  and  $CH_4$  lifetime, we perform targeted simulations with the EMAC model in which either one or the other inventory is prescribed (see Table 1). In the following sections, the EMAC model, the simulation set-up including information about the emission inventories, and the tagging method as well as the calculation of the  $CH_4$  lifetime are described. In Sect. 2.4 we also describe the two methods to estimate the contribution of individual tagging categories to the overall change of the  $CH_4$  lifetime.

#### 2.1 Model description and simulation strategy

We perform simulations with the global chemistry-climate model EMAC (ECHAM/MESSy Atmospheric Chemistry). The EMAC model is composed of the European Center Hamburg Model version 5 (ECHAM5; Roeckner et al., 2006) as the dynamical base model and the Modular Earth Submodel System (MESSy; Jöckel et al., 2010). MESSy describes meteorological processes and atmospheric chemistry in a modular framework (Jöckel et al., 2010). The MESSy approach flexibly combines individual submodels with a base model (Jöckel et al., 2005). These submodels describe processes in the troposphere and middle atmosphere and their interaction with oceans, land, and anthropogenic influences (Jöckel et al., 2010).

In this study, EMAC was applied in T42L90MA resolution, i.e. at a triangular (T) truncation at wave number 42 of the spectral dynamical core, corresponding to a quadratic Gaussian grid of approximately  $2.8^{\circ} \times 2.8^{\circ}$  resolution in latitude and longitude and 90 vertical levels, with the uppermost level centered around 0.01 hPa ( $\approx 80 \text{ km}$  height) in the middle atmosphere (MA).

Nudging by Newtonian relaxation towards ECMWF ERA-5 reanalysis data (Hersbach et al., 2020) was performed aiming at a realistic representation of the meteorological situation at a given time. Nudged are the temperature, divergence, vorticity, and the logarithm of surface pressure (Jöckel et al., 2016). Wave zero of the temperature (e.g. the mean temperature) is not nudged.






To decouple dynamics from chemistry, EMAC was operated in quasi chemistry-transport model (QCTM) mode (Deckert et al., 2011). In this mode, mixing ratios of radiatively active constituents are prescribed for the radiation calculations, and the hydrological cycle is decoupled from the chemistry. This assures that differences of the emission inventories prescribed in the simulations do not affect the model's meteorology, not even numerically. Accordingly, differences of the chemical decomposition between the simulations occur solely due to differences in the emissions and prescribed lower boundary mixing ratios of CH<sub>4</sub> and GHGs.

For the calculation of lighting  $NO_x$  emissions we use the parameterization by Grewe et al. (2001). Biogenic  $C_5H_8$  emissions and soil  $NO_x$  emissions are calculated with the submodel ONEMIS (Kerkweg et al., 2006) applying the parameterization of Guenther et al. (1995) and Yienger and Levy II (1995), respectively. Due to the identically simulated meteorology in all simulations due to the QCTM mode, these online calculated, meteorology dependent emissions are exactly the same in both simulations with no differences, not even numerical noise. In addition to the online calculated natural emissions, a climatology of biogenic emissions of NMHCs and CO is prescribed from the Global Emissions InitiAtive (GEIA) in all simulations.

Overall, we performed three different simulations as listed in Table 1, EMIS-01, EMIS-02 and EMIS-01-fODS. The simulation set-up is similar to the set-up of the CCMI-2022 REFD1SD simulation (hindcast with specified dynamics), but we deviate from the CCMI naming convention to clarify that we have performed the simulations specifically for this publication, and that they are not identical to the simulations originally performed for CCMI-1 and CCMI-2022. The simulations EMIS-01 and EMIS-02 differ between the used emission inventories for shorter lived chemical species (i.e.  $NO_x$ , CO,  $SO_2$ ,  $NH_3$ , and VOCs, see Sect. 2.2 for more details), the prescribed lower boundary mixing ratios of GHGs (in particular also  $CH_4$ ), and the prescribed lower boundary mixing ratios of ozone depleting substances (ODS) (see Jöckel et al., 2016, for more details). The global mean surface  $CH_4$  mixing ratio is about 0.01 ppm larger in the simulation EMIS-02 compared to EMIS-01 (see Fig. S12 in the supplement), which is expected to cause a small extension of the  $CH_4$  lifetime, which counteracts the overall shortening of the  $CH_4$  lifetime. The prescribed lower boundary for  $N_2O$  is nearly identical, which is also reflected by the total mass of  $N_2O$  (see Fig. S13 in the supplement). The influence of the change in the prescribed ODS turned out to have no influence on the tropospheric  $CH_4$  lifetime and only a small influence on tropospheric  $O_3$  (see Fig. S2 in the supplement). Therefore, only the simulations EMIS-01 and EMIS-02 are further analysed in our study.

We investigated the period 2000 - 2010, with a spin-up phase for the simulations comprising 1998 and 1999. The model output is originally output in netCDF format with global data coverage as snapshots every 10 h of simulated time on model levels. Based on this, monthly averages are calculated on pressure levels.

| Simulation name | ozone precursor emissions | GHG lower boundary condition (incl. methane) | ODS       |
|-----------------|---------------------------|----------------------------------------------|-----------|
| EMIS-01         | CCMI-1                    | CCMI-1                                       | CCMI-1    |
| EMIS-01-ODS     | CCMI-1                    | CCMI-1                                       | CCMI-2022 |
| EMIS-02         | CCMI-2022                 | CCMI-2022                                    | CCMI-2022 |

**Table 1.** Overview of the performed numerical experiments.

# 2.2 Prescribed emission inventories of ozone precursor species

The emission inventory for CCMI-1 is based on the MACCity emission inventory (Granier et al., 2011), an extension of the Atmospheric Chemistry and Climate Model Intercomparison Project (ACCMIP) emissions dataset based on 1960 and 2000, as well as 2005 and 2010 RCP 8.5 emissions (CCMI-1 simulation protocol). The emission inventory for CCMI-1 contains anthropogenic emissions based on the product of estimates for activity data from country records and international organizations and emission factors (Granier et al., 2011). The basis for global biomass burning emissions are satellite data (Granier et al., 2011). The MACCity inventory includes global emissions of CO, NO<sub>x</sub> and NMHCs, whereby it considers a monthly resolved seasonal cycle (Granier et al., 2011). The ACCMIP dataset according to Lamarque et al. (2005) was expanded and calculated for emissions from biomass burning on the basis of the annual monthly average trace gas emissions from the spatially and temporally modified RETRO (REanalysis of the TROpospheric chemical composition; Schultz et al., 2007) carbon emissions data from GFED-v2 (Global Fire Emissions Database; van der Werf et al., 2006) for 1997 to 2008 (Granier et al., 2011). A
detailed description of the implementation of the emission inventory for EMAC is provided in the supplement of Jöckel et al. (2016).

The basis of the emission inventory for CCMI-2022 is the Community Emissions Data System (CEDS) dataset, which includes annual historical (1750-2014) anthropogenic emissions for the species CO,  $NO_x$ , and NMHCs (Hoesly et al., 2018). The dataset includes existing energy consumption data, and regional and country emission inventories. The dataset provides annual emissions at country and sector level with monthly seasonal variations. The biomass burning emissions in CCMI-2022 are according to van Marle et al. (2017).

Time series of the years 2000 - 2010 of global emissions of  $NO_x$ , NMHCs and CO for individual sectors anthropogenic non-traffic, biogenic, land transport, shipping, biomass burning and aviation, as prescribed in the simulations EMIS-01 or EMIS-02 can be found in the supplement (Figures S8, S9 and S10). Global anthropogenic  $NO_x$  emissions are larger by 9.02 Tg(N) year<sup>-1</sup> on average for the period 2000 - 2010 for the simulation EMIS-02 compared to EMIS-01 (see Tab. 3). In the NH, the sectors anthropogenic non-traffic, land transport and shipping contribute most to the difference in total  $NO_x$  emissions, with  $NO_x$  emissions being larger in the EMIS-02 simulation than in EMIS-01. In the SH, the largest contributors to the  $NO_x$  emission difference are the sectors biomass burning and shipping (see Tab. 3). The difference of  $NO_x$  emissions from shipping between EMIS-01 and EMIS-02 peaks in the year 2008 (see Fig. S8 in the supplement), as the emissions prescribed for EMIS-02 account for the increase of shipping emissions until 2008 and the decline after 2008 (Hoesly et al., 2018).

Global CO emissions are reduced by 28.68 Tg(C) year<sup>-1</sup> on average for the period 2000 - 2010 for EMIS-02 compared to EMIS-01 (see Tab. 3). The reduction is dominated by the biomass burning sector in both hemispheres, whereas CO emissions from the anthropogenic non-traffic and land transport sectors are larger in EMIS-02 (see Tab. 3).

Global NMHC emissions are enhanced by 56.90 Tg(C) year<sup>-1</sup> on average for the period 2000 - 2010 for EMIS-02 (see Tab. 3). The increase is dominated by the anthropogenic non-traffic sector, followed by biomass burning and land transport in the NH. In the SH, differences in the anthropogenic non-traffic and biomass burning sectors contribute most as well.

# 2.3 TAGGING for source attribution






To understand the effects of the emission changes on  $O_3$  precursors,  $O_3$  and OH in more detail, we apply the TAGGING submodel as described in detail by Grewe et al. (2017); Rieger et al. (2018). The TAGGING submodel, as technical implementation of the tagging approach (a labeling technique) is used to diagnose contributions of individual source categories (emission sectors and/or regions or other source processes) to the mixing ratio of tropospheric  $O_3$  and its precursors (Grewe et al., 2010). For this purpose, the TAGGING submodel distributes the calculated chemical production rates among additional, so called tagged tracers (Grewe et al., 2012). Each tagged tracer represents the contribution of one source category (emission sector and/or region or other process) to a specific species. The chemical production and loss rates for individual chemical reactions are provided by the submodel Module Efficiently Calculating the Chemistry of the Atmosphere (MECCA; Sander et al., 2019) (Grewe et al., 2017), which solves the non-linear kinetic reaction system mathematically described by an ordinary differential equation system. The distribution of the production and loss rates among the tagged tracers follows a combinatorial approach based on the concentrations of the tagged species.

According to Grewe et al. (2017), tagged species are  $NO_y$ , NMHCs, CO, PAN,  $O_3$ ,  $HO_2$ , and OH. According to Rieger et al. (2018), OH and  $HO_2$  are treated with a steady-state approach. According to Grewe et al. (2017), "all possible combinations between a tagged  $NO_x$  species and another tagged  $HO_x$  species are evaluated and their probability is calculated in accordance with the chemical production rate calculation". It is important to note that the chosen approach intrinsically takes into account the non-linear nature of the kinetic system. Details are documented by Grewe et al. (2017). In this study, we use the TAGGING submodel v1.1 according to Rieger et al. (2018), which was updated to be consistent with the used chemical mechanism. Thereby, the changed partitioning of the  $O_3$  production rates is the most important adjustment, whereas the TAGGING for OH and  $HO_2$  remained unchanged (see Stecher, 2024, Chapter 3.1.6 for more details).

In this study, we investigate ten source categories (also referred to as tagging categories in the further course of this text) and their contribution to tropospheric  $O_3$ , OH, and the  $CH_4$  lifetime. The latter is derived from the tagged OH as explained in Sect. 2.4 in more detail. We use the same definition of tagging categories as Grewe et al. (2017), distinguishing between natural and anthropogenic sources. The definition of the ten tagging categories is listed in Table 2. The term category is used independently of the type of source, e.g. emission, decomposition process, or production in the stratosphere. In contrast, we use the term (emission) sector only for emissions. For example, the category land transport represents the contribution of the land transport emission sector, whereas the category stratosphere represents downward transport of  $O_3$  formed in the stratosphere, which is not related to any emission sector.

**Table 2.** Overview of the categories for the tagging source attribution applied in this study. Detailed definitions of the categories are given by Grewe et al. (2017). The anthropogenic non-traffic category contains all non-traffic related activities i.e. industry, households, energy, and agriculture including agricultural waste burning

| name                      | abbreviation | anthropogenic/natural | description                                                  |  |  |  |
|---------------------------|--------------|-----------------------|--------------------------------------------------------------|--|--|--|
| anthropogenic non-traffic | IND          | anthropogenic         | anthropogenic emissions not related to transport             |  |  |  |
| land transport            | TRA          | anthropogenic         | land transport (IPCC codes 1A3b_c_e) emissions               |  |  |  |
| shipping                  | SHP          | anthropogenic         | Global shipping emissions (IPCC code 1A3d)                   |  |  |  |
| aviation                  | AIR          | anthropogenic         | Global aviation emissions                                    |  |  |  |
| stratosphere              | STR          |                       | downward transport from the stratosphere (more specifically, |  |  |  |
|                           |              |                       | production of ozone by O <sub>2</sub> )                      |  |  |  |
| biogenic                  | SOIL         | natural               | biogenic emissions and soil- $\mathrm{NO}_{\mathrm{x}}$      |  |  |  |
| biomass burning           | BIO          | anthropogenic         | emissions from biomass burning                               |  |  |  |
| CH4                       | CH4          | both                  | degradation of CH <sub>4</sub> as source of NMHCs            |  |  |  |
| N2O                       | N2O          | both                  | degradation of $N_2O$ as source of $NO_y$                    |  |  |  |
| lightning                 | LIG          | natural               | emissions of lightning $\mathrm{NO}_{\mathrm{x}}$            |  |  |  |

# 200 2.4 Calculation of methane lifetime

In this study, we calculate the (tropospheric)  $\mathrm{CH_4}$  lifetime with respect to the oxidation with OH according to Jöckel et al. (2016) as

$$\tau_{CH_4,OH_{total}} = \frac{\sum_{b \in B} m_{CH_4}}{\sum_{b \in B} k_{CH_4 + OH}(T) \cdot c_{air}(T, p, q) \cdot OH \cdot m_{CH_4}},$$
(1)

with  $m_{CH_4}$  being the mass of CH<sub>4</sub> in [kg],  $k_{CH_4+OH}(T)$  the temperature T dependent reaction rate coefficient of the reaction CH<sub>4</sub> + OH  $\rightarrow$  products in [cm<sup>3</sup> s<sup>-1</sup>],  $c_{air}$  the concentration of air in [cm<sup>-3</sup>] and OH the mole fraction of OH in [mol mol<sup>-1</sup>] in all grid boxes b ∈ B. B is the region, for which the lifetime should be calculated, e.g. all grid boxes below the tropopause for the mean tropospheric lifetime, and all grid boxes below the tropopause in each hemisphere to derive a hemispheric CH<sub>4</sub> lifetime (denoted as SH and NH for the Southern and Northern Hemispheres, respectively). For the CH<sub>4</sub> lifetime calculation a climatological tropopause, defined as tp<sub>clim</sub>= 300 hPa - 215 hPa  $\cdot$  cos<sup>2</sup>( $\phi$ ), with  $\phi$  being the latitude in degrees north, is used as recommended by Lawrence et al. (2001).

In analogy, we calculate the CH<sub>4</sub>-weighted OH (Lawrence et al., 2001) as

$$OH_{weighted} = \frac{\sum_{b \in B} k_{CH_4 + OH}(T) \cdot c_{air}(T, p, q) \cdot OH \cdot m_{CH_4}}{\sum_{b \in B} k_{CH_4 + OH}(T) \cdot m_{CH_4}}.$$
(2)

The  $CH_4$  lifetime corresponding to the oxidation with OH of one tagging category is calculated by substituting the total OH in Eq. 1 by the TAGGING tracer  $OH_i$ , e.g.  $OH_{tra}$ .

$$\tau_{CH_4,OH_i} = \frac{\sum\limits_{b \in B} m_{CH_4}}{\sum\limits_{b \in B} k_{CH_4+OH}(T) \cdot c_{air}(T,p,q) \cdot OH_i \cdot m_{CH_4}}.$$
 (3)

Note that for adding the  $\mathrm{CH_4}$  lifetimes of multiple categories the inverse of the sum of the reciprocals of individual  $\mathrm{CH_4}$  lifetimes needs to be calculated. For instance, the  $\mathrm{CH_4}$  lifetime with respect to the combined  $\mathrm{OH}$  loss of all categories is calculated as

$$\tau_{CH_4,OH_{sum}} = \left(\sum_{i} \frac{1}{\tau_{CH_4,OH_i}}\right)^{-1} \\
= \left(\frac{1}{\tau_{CH_4,OH_{ind}}} + \frac{1}{\tau_{CH_4,OH_{shn}}} + \frac{1}{\tau_{CH_4,OH_{bio}}} + \frac{1}{\tau_{CH_4,OH_{sin}}} + \frac{1}{\tau_{CH_4,OH_{sin}}} + \frac{1}{\tau_{CH_4,OH_{sin}}} + \frac{1}{\tau_{CH_4,OH_{sin}}} + \dots\right)^{-1}.$$
(4)

Since not all source and loss reactions of OH, but a reduced reaction system, are considered by the TAGGING (Rieger et al., 2018), the sum of all OH TAGGING tracers can deviate from the total tracer OH. To quantify the deviation, we calculate the residuum lifetime, i.e. the CH<sub>4</sub> lifetime corresponding to CH<sub>4</sub> loss with the portion of OH that is unaccounted for by the TAGGING as

$$\tau_{CH_4,OH_{res}} = \left(\frac{1}{\tau_{CH_4,OH_{total}}} - \frac{1}{\tau_{CH_4,OH_{sum}}}\right)^{-1} = \left(\frac{1}{\tau_{CH_4,OH_{total}}} - \sum_{i} \frac{1}{\tau_{CH_4,OH_i}}\right)^{-1}.$$
 (5)

The difference between  $\tau_{CH_4,OH_{total}}$  and  $\tau_{CH_4,OH_{sum}}$  excluding the residuum lifetime is about 0.1 years for both simulations. The mean values and the respective standard deviations (due to interannual variations) of  $\tau_{CH_4,OH_{res}}$  are 398.41  $\pm$  40.99 years for EMIS-01 in the NH, 378.74  $\pm$  89.30 years for EMIS-02 in the NH, 434.54  $\pm$  49.52 years for EMIS-01 in the SH, and 339.56  $\pm$  38.97 years for EMIS-02 in the SH, which is notably larger than the respective estimates of the other categories. Since the inverse of  $\tau_{CH_4,OH_{res}}$  determines the influence on the CH<sub>4</sub> lifetime with respect to total OH (Eq. 4), we conclude that the CH<sub>4</sub> loss via oxidation with OH that is unaccounted by the TAGGING is sufficiently small. In particular, it is smaller than the contribution of any of the tagging categories. Note that the CH<sub>4</sub> lifetime with respect to total OH,  $\tau_{CH_4,OH_{total}}$ , and the lifetime summed over all tagging categories including  $\tau_{CH_4,OH_{res}}$ ,  $\tau_{CH_4,OH_{sum}}^*$ , are identical, so that CH<sub>4</sub> loss by the total OH tracer is accounted for when  $\tau_{CH_4,OH_{res}}$  is included in Eq. 4.

In the results section, we attribute the change of the  $\mathrm{CH}_4$  lifetime with respect to total OH between the two simulations to the individual tagging categories. Therefore we calculate  $\Delta \tau_{CH_4,OH_i}$ , which represents the lifetime reduction attributed to an individual source category i. Note that  $\Delta \tau_{CH_4,OH_i}$  is not just given by the difference of  $\mathrm{CH}_4$  lifetimes of the category i between the two simulations, i.e.  $\Delta \tau_{CH_4,OH_i} \neq (\tau_{CH_4,OH_i}(\mathrm{EMIS-02}) - \tau_{CH_4,OH_i}(\mathrm{EMIS-01}))$ , because of the fact that the reciprocals of individual  $\mathrm{CH}_4$  lifetimes need to be added when the sum of individual categories is calculated. We calculate  $\Delta \tau_{CH_4,OH_i}$  with the following two methods, whose results are compared.

The derivation of the first method is shown in Appendix A. Briefly, we express the lifetime with respect to total OH  $\tau_{CH_4,OH_{total}}$  (Eq. 1) in dependence of the lifetime of the category of interest  $\tau_{CH_4,OH_i}$ , for which we calculate the derivative

with respect to  $\tau_{CH_4,OH_i}$  (see Eq. A2 and A3 in Appendix A). Then, we approximate the derivative with the finite change of CH<sub>4</sub> lifetime (see Eq. A4 in Appendix A), which results in the following relation for the lifetime reduction attributed to category i

$$\Delta \tau_{CH_4,OH_i} = \frac{(\tau_{CH_4,OH_{total}}(\text{EMIS-01}))^2}{(\tau_{CH_4,OH_i}(\text{EMIS-01}))^2} \cdot (\tau_{CH_4,OH_i}(\text{EMIS-02}) - \tau_{CH_4,OH_i}(\text{EMIS-01})).$$
 (6)

For example, using the first method,  $\Delta \tau_{CH_4,OH_{tra}}$ , which represents the contribution to the overall lifetime change attributed to the change of land transport emissions (CCMI-2022 vs. CCMI-1), is calculated as

$$\Delta \tau_{CH_4,OH_{tra}}^{M1} = \frac{\tau_{CH_4,OH_{total}}^2(\text{EMIS-01})}{\tau_{CH_4,tra}^2(\text{EMIS-01})} \cdot (\tau_{CH_4,OH_{tra}}(\text{EMIS-02}) - \tau_{CH_4,OH_{tra}}(\text{EMIS-01})). \tag{7}$$

For the second method, we exchange the  $CH_4$  lifetime of the category of interest,  $\tau_{CH_4,OH_i}$ , with the  $CH_4$  lifetime of this category derived from the other simulation in Eq. 4. Thus, with the second method the contribution of one individual category, e.g. changed land transport emissions, to the overall lifetime change is calculated as

$$\Delta \tau_{CH_4,tra}^{M2} = \left(\frac{1}{\tau_{CH_4,OH_{sum}}^*(\text{EMIS-01})} - \frac{1}{\tau_{CH_4,OH_{tra}}^{exchanged}}\right)^{-1},\tag{8}$$

where  $au_{CH_4,tra}^{exchanged}$  is calculated as

$$\tau_{CH_4,OHtra}^{exchanged} = \left(\frac{1}{\tau_{CH_4,OH_{tra}}(\text{EMIS-02})} + \sum_{i \neq tra} \frac{1}{\tau_{CH_4,OH_i}(\text{EMIS-01})}\right)^{-1}.$$
 (9)

#### 255 3 Results



# 3.1 Contribution to tropospheric ozone

In this section, we show the difference of total  $O_3$  between the simulations EMIS-01 and EMIS-02, as well as the change of the contributions of individual tagging categories. The tropospheric  $O_3$  columns (Fig. 2) of the simulations EMIS-01 and EMIS-02 in Dobson Units (DU), as well as the relative differences between both simulations in percent, are shown for the period 2000 – 2010. The prognostic tropopause as diagnosed by the model is used as boundary between troposphere and stratosphere. The relative differences are calculated with the EMIS-01 results as the reference. The patterns of the tropospheric  $O_3$  columns show the seasonal cycle of  $O_3$ , i.e. more  $O_3$  is present in NH summer than in NH winter. In the NH, EMIS-02 ( $\bar{x} = 38.88$  DU; STD = 0.59) shows a 1.39 DU (= 3.7%) larger tropospheric  $O_3$  column compared to EMIS-01 ( $\bar{x} = 37.49$  DU; STD = 0.05) for the period 2000 - 2010. In the SH, EMIS-02 ( $\bar{x} = 27.90$  DU; STD = 0.43) also shows larger values than EMIS-01 ( $\bar{x} = 26.71$  DU; STD = 0.32). The absolute difference averaged over the period 2000 - 2010 is 1.19 DU in the SH, which corresponds to 4.5%. The tropospheric  $O_3$  column is especially large in the NH between 20° and 50°, showing values up to 60 DU. Within the period 2000 - 2010, the tropospheric  $O_3$  column in the EMIS-02 simulation between 10°S and 40° N is up to 12% higher than in the EMIS-01 simulation. The maximum difference of the tropospheric  $O_3$  column occurs in the years 2008 and 2009. Globally

Figure 2. Tropospheric  $O_3$  columns (DU) of the EMIS-01 and EMIS-02 simulation results and the relative differences with the EMIS-01 results as the reference. The data are monthly averaged for the period 2000 - 2010.

and temporally averaged, the simulation with the EMIS-02 emissions shows a 4% larger tropospheric  $O_3$  column compared to 270 EMIS-01.

The zonal mean of the total  $O_3$  volume mixing ratio in nmol/mol and the relative differences between EMIS-01 and EMIS-02 are shown in Fig. S1 in the supplement.  $O_3$  mixing ratios are larger in EMIS-02 compared to EMIS-01 throughout the troposphere with a maximum relative difference in the lower levels in the tropics. Fig. S3 shows that effective production of  $O_3$  is enhanced in EMIS-02 most strongly in lower levels.  $O_3$  loss is enhanced as well, however less strongly, so that the net effect is to enhance  $O_3$  concentration.

Figure 3 shows the relative zonal mean contribution of the categories shipping, anthropogenic non-traffic, land transport, aviation and biomass burning to total  $O_3$ , as well as the difference of the contribution between the two simulations. A corresponding plot for the categories  $N_2O$ , lightning, stratosphere,  $CH_4$  and biogenic can be found in the supplement (Fig. S5). The differences between the  $O_3$  contributions are a result of the complex interplay between the emission changes and the background chemistry. In the NH, especially land transport, anthropogenic non-traffic and shipping emissions are larger in EMIS-02 compared to EMIS-01 (see Sect. 2.2). However, only for land transport and anthropogenic non-traffic emissions the contribution to  $O_3$  are larger in EMIS-02 compared to EMIS-01 in the NH. For shipping emissions, the differences between EMIS-02 and EMIS-01 are not significant over most regions in the NH. The increase of the contributions from land transport and anthropogenic non-traffic emissions lead to a decrease of contributions from biomass burning in the NH in EMIS-02 compared to EMIS-01, even though the emissions are only slightly lower in EMIS-02 compared to EMIS-01. In the SH, however, the strong increase of the biomass burning emissions in EMIS-02 lead to larger contributions in EMIS-02 compared to EMIS-01. The contribution of aviation emissions to  $O_3$  is lower in EMIS-02 compared to EMIS-01, partly due to the slightly lower emissions, but also due to the artificially shifted aviation emissions from the tropics to the polar regions in the CEDS emission inventory as reported by Thor et al. (2023).


Figure 3. Relative contributions to total  $O_3$  of the tagging categories (a) shipping, (b) anthropogenic non-traffic, (c) land transport (d) aviation and (e) biomass burning. The left and middle columns show the contribution of the respective category in the simulations EMIS-01 and EMIS-02, respectively. The right column shows the differences of the relative contributions to  $O_3$  between the two simulation in percentage points. Zonal means of the years 2000 - 2010 are presented. Hatches in the delta plot indicate a p-value  $\geq 0.05$  from the dependent t-test for paired samples.

To understand the effects of the changed emissions onto  $O_3$  production in more detail, we calculate the  $O_3$  burden efficiency with respect to  $NO_x$  emissions (see also Mertens et al., 2024), which is defined for example for the shipping category as:

$$\chi^{SHP} = \frac{B(O_3^{SHP})}{E(NO_s^{SHP})}.$$
 (10)

In this definition,  $B(O_3^{\rm SHP})$  is the annual mean total atmospheric mass of  $O_3$  attributed to shipping emissions (in kg), while  $E(NO_x^{\rm SHP})$  are the global annual mean shipping emissions of  $NO_x$  (in kg(N) per year).

Figure 4 shows the burden efficiency for the tagging categories, for which  $NO_x$  emissions are either prescribed or calculated online except for the category aviation. The aviation category is shown separately in Fig. S14 because it has a much larger burden efficiency (Mertens et al., 2024). The burden efficiency of the aviation category,  $\chi^{AIR}$ , is larger in the simulation EMIS-02. The maximum difference of  $\chi^{AIR}$  between the two simulations is about 10% in the year 2010 as  $\chi^{AIR}$  of the simulation EMIS-01 decreases over the 11 analyzed years. This is likely due to the different geographical distribution of the aviation emissions as explained above.

For the categories anthropogenic non-traffic and shipping, the burden efficiency is roughly the same in both simulations indicating that the larger prescribed  $NO_x$  emissions result in a correspondingly large increase of the  $O_3$  production. For the categories biogenic, land transport and biomass burning the burden efficiency is smaller in the simulation EMIS-02 compared to EMIS-01. For the category biogenic, this implies a reduced  $O_3$  production from identical biogenic soil  $NO_x$  emissions as the online calculated emissions are identical per set-up definition (see Sect. 2.1). For the categories land transport and biomass burning, the  $NO_x$  emissions, as well as the  $O_3$  burden, are enhanced in the simulation EMIS-02. However, the emissions and the  $O_3$  burden do not increase uniformly with each other, due to the non-linearity of the  $O_3$  chemistry. For the category lightning on the contrary, the burden efficiency is larger in the simulation EMIS-02 compared to EMIS-01, likely due to the differences in the aviation emissions. The lightning  $NO_x$  emissions are identical in the two simulations, but  $O_3$  is produced more efficiently in the simulation EMIS-02.

#### 3.2 Contribution to OH and methane lifetime

The results of the analyses of the relative contribution to OH and the associated  $CH_4$  lifetime of the individual categories are presented in this section.

#### 3.2.1 Contribution to OH





Figure 5 shows the zonal mean mixing ratios of OH in both simulations, as well as their difference. The largest differences are located around the tropical tropopause, where the tropospheric OH mixing ratio is largest, and shows larger OH mixing ratios in the simulation EMIS-02 compared to EMIS-01. The OH mixing ratios are also larger in the lower (up to about 500 hPa) tropical troposphere in EMIS-02, but smaller between 500 and 300 hPa. To complement Fig. 5, Fig. S6 in the supplement shows the zonal mean number concentration of OH weighted by the reaction with CH<sub>4</sub> (see Eq. 2). The zonal mean distribution of weighted OH is similar in both simulations with maxima in the tropical tropopause region and in the lower tropical troposphere.

**Figure 4.** Global burden efficiency  $\chi^i$  of the categories anthropogenic non-traffic, land transport, shipping, biomass burning, biogenic and lightning (in [kg(O<sub>3</sub>) / kg(N)]).

The OH concentration is larger in EMIS-02 compared to EMIS-01 in both of these regions, consistent with the difference of OH mixing ratios shown in Fig. 5.

Figure 6 shows the tropospheric mean OH number concentration of individual tagging categories weighted by the reaction with  $CH_4$ . Anthropogenic emissions of  $O_3$  precursors contribute importantly to OH. The categories anthropogenic non-traffic, land transport, shipping and aviation combined represent 36% of the global tropospheric weighted OH for EMIS-01, and 39% for EMIS-02. Their role is more important in the NH compared to the SH. Biomass burning emissions contribute 5% to the global tropospheric weighted OH in EMIS-01, and 6% in EMIS-02. Emissions of lightning  $NO_x$  contribute about 25% to the global tropospheric OH, biogenic emissions about 12%, transport of  $O_3$  formed in the stratosphere about 9%, and decomposition of  $CH_4$  and  $N_2O$  about 7% and 4%, respectively. The contributions of natural emissions, decomposition processes and transport from the stratosphere are more important in the SH compared to the NH. In particular, emissions of lightning  $NO_x$  account for about 30% in the SH, and only about 20% in the NH.




The global tropospheric weighted OH is enhanced in the categories anthropogenic non-traffic ( $+0.03 \times 10^6$  molec. cm<sup>-3</sup>), land transport ( $+0.01 \times 10^6$  molec. cm<sup>-3</sup>), shipping ( $+0.02 \times 10^6$  molec. cm<sup>-3</sup>) and biomass burning ( $+0.02 \times 10^6$  molec. cm<sup>-3</sup>) in EMIS-02 compared to EMIS-01, whereas it is reduced by  $-0.01 \times 10^6$  molec. cm<sup>-3</sup> in the categories aviation and biogenic. The global tropospheric OH attributed to the categories stratosphere,  $N_2O$ ,  $CH_4$ , and lightning remains unchanged in both simulations.

Figure 7 shows the zonal mean contribution of the categories shipping, anthropogenic non-traffic, land transport, aviation and biomass burning to total OH, as well as the difference of the contribution between the two simulations. A corresponding plot for the categories  $N_2O$ , lightning, stratosphere,  $CH_4$  and biogenic can be found in the supplement (Fig. S7). Zonal means of the year 2000 - 2010 are presented. The OH contributions from the shipping category show maxima in the lower troposphere in the NH in EMIS-01 and in EMIS-02 in the lower troposphere in the NH and SH. The OH contributions from anthropogenic non-traffic and land transport categories show maxima in the lower troposphere in the NH. The aviation category has its maximum in the upper troposphere and above the tropopause in the NH. In the category biomass burning, the maximum is at the equator in the lower troposphere and in the tropical tropopause region in both simulations. In the simulation EMIS-01, the contribution from the biomass burning category shows another maximum in the northern mid to high latitudes.

The panels in the right column show the differences of the relative contributions of one category to total OH in percentage points between the two simulations. The differences indicate that tropospheric OH contributions are not the same for EMIS-01 and EMIS-02. The difference is predominantly positive in all categories, indicating larger contributions to total OH from these categories in EMIS-02 compared to EMIS-01, except for the aviation category. In the shipping category, the largest increases in relative OH mixing ratio contribution (10 percentage points) between EMIS-01 and EMIS-02 can be located in the NH and SH up to 900 hPa. The contribution of the shipping category is reduced by up to -4 percentage points in a small area in the NH (around 40°N, 1000 hPa). The contribution of the anthropogenic non-traffic category to total OH is overall increased with a maximum difference of 4 percentage points around 50°N close to the surface. This category also shows reduced contributions (up to -3 percentage points) in a small area in the NH (40° – 70°N, 1000 hPa). For the land transport category, the difference (6 percentage points) between the two simulations is most pronounced between 60° and 80°N at 1000 hPa. The land transport category also shows reduced contributions (-2 percentage point) in the SH (20° - 80°S, 700 - 1000 hPa). The contribution of the aviation category is increased by up to 2 percentage points at the tropopause between 70°N and 90°N. This category also shows reduced contributions (-2 percentage point) between 10°N and 40°N between 250 and 700 hPa. For the biomass burning category, the largest increase in the contribution (4 percentage points) occurs between 0° and 15°S, at 750 – 950 hPa. The biomass burning category also shows reduced contributions (-7 percentage points) in the NH (around 50° – 70°N, 700 – 900 hPa).

The categories shipping, anthropogenic non-traffic, land transport, aviation and biomass burning show the largest positive and the largest negative differences between EMIS-01 and EMIS-02 with respect to the contribution to OH. The other categories show lower positive and/or lower negative changes in the contribution to total OH (see Fig. S7 in the supplement).

# 3.2.2 Contribution to methane lifetime







In this section, we attribute the difference of the tropospheric  $CH_4$  lifetime with respect to the oxidation with OH in the simulations EMIS-01 and EMIS-02 to individual tagging categories. We show results for the NH and SH separately. The black bars in Figure 8 show the  $CH_4$  lifetime with respect to the oxidation with total OH. In the NH, the absolute difference of the  $CH_4$  lifetime with respect to total OH between the simulations EMIS-01 (7.86 years) and EMIS-02 (7.61 years) is -0.25 years,

Figure 5. Multi-annual (2000 - 2010), zonally averaged OH volume mixing ratio  $[10^{-15} \text{mol mol}^{-1}]$  and the absolute differences between EMIS-01 and EMIS-02 simulation results.

Figure 6. Tropospheric OH number concentration of individual tagging categories weighted by the reaction with CH<sub>4</sub> calculated as  $[OH]_{weighted} = \frac{\sum (k_{CH_4,OH} \cdot M_{CH_4} \cdot [OH])}{\sum (k_{CH_4,OH} \cdot M_{CH_4})}$  following Lawrence et al. (2001) (in 10<sup>6</sup> molecules cm-3). The first two bars show global means corresponding to the simulations EMIS-01 and EMIS-02, respectively. The last four bars show results for the Northern Hemisphere (NH) or Southern Hemisphere (SH) separately.

which corresponds to -3.2%. In the SH, the CH<sub>4</sub> lifetime of the simulation EMIS-01 (9.40 years) and EMIS-02 (9.00 years) differs by -0.4 years, which corresponds to -4.3%.

Figure 7. Relative contributions to total OH of the tagging categories (a) shipping, (b) anthropogenic non-traffic, (c) land transport (d) aviation and (e) biomass burning. The left and middle columns show the contribution of the respective category in the simulations EMIS-01 and EMIS-02, respectively. The right column shows the differences of the relative contributions to OH between the two simulation in percentage points. Zonal means of the years 2000 - 2010 are presented. Hatches in the delta plot indicate a p-value  $\geq 0.05$  from the dependent t-test for paired samples.

We calculate the contribution of each category to the  $CH_4$  lifetime change with respect to total OH using the two methods, which are explained in Sect. 2.4. The results are shown by the blue and red bars, respectively, in Fig. 8. The relative contributions are consistent for both methods. However, method 1 indicates overall less pronounced  $CH_4$  lifetime changes for the individual categories and underestimates the  $CH_4$  lifetime change with respect to total OH. The sum of all individual contributions calculated using method 2 reproduces the  $CH_4$  lifetime change with respect to total OH well in the NH, and slightly overestimates it in the SH.

Our results show that the anthropogenic non-traffic, land transport, biomass burning, and shipping categories contribute most to the  $CH_4$  lifetime reduction from EMIS-01 to EMIS-02 in the NH. In the SH, also the anthropogenic non-traffic, biomass burning and shipping categories contribute to the lifetime reduction. However, the results indicate that the land traffic category leads to a slight extension of  $CH_4$  lifetime in the SH. The impact of the categories  $N_2O$ , lightning, stratosphere and  $CH_4$  to the  $CH_4$  lifetime difference between the two simulations is minor. The aviation category leads to an increase of the  $CH_4$  lifetime in the NH by 0.06 years. The biogenic category leads to an extension of the  $CH_4$  lifetime in both hemispheres.

# 3.3 Synthesis



We have performed targeted simulations to assess the impact of the changed emission inventories of  $O_3$  precursors (CO, NMHCs,  $NO_x$ ) from CCMI-1 to CCMI-2022 over the years 2000 - 2010. We find that the changed emission inventories lead to an increase of the tropospheric  $O_3$  column of 3.7%, and to a shortening of the tropospheric  $CH_4$  lifetime with respect to OH by 0.25 years in the NH, which corresponds to 3.2%. In the SH, the tropospheric  $O_3$  column increases by 4.5%, and the tropospheric  $CH_4$  lifetime shortens by 0.4 years, which corresponds to 4.3%. We further use the TAGGING submodel to estimate the contribution of individual tagging categories to the changes of total  $O_3$ , total OH, and  $CH_4$  lifetime with respect to total OH. In the following, we summarize the categories that contribute most to the  $O_3$  and  $CH_4$  lifetime differences between the two simulations, and compare our results to the changes of prescribed  $O_3$  precursor emissions in the corresponding sectors.

In the NH, the largest changes of both, the tropospheric  $O_3$  column and the  $CH_4$  lifetime, are attributed to the anthropogenic non-traffic and land transport categories, which correspond to the sectors with the largest change of  $NO_x$  emissions between CCMI-1 and CCMI-2022 (see Tab. 3).  $NO_x$  emissions from biomass burning are slightly lower by -0.21 Tg(N) in CCMI-2022. This sector contributes nevertheless to a shortening of the  $CH_4$  lifetime, which might be explained by the comparable strong reduction (CCMI-2022 compared to CCMI-1) of CO emissions, resulting in an enhanced abundance of OH, as the most important sink of CO is the oxidation with OH (e.g., Nguyen et al., 2020). The corresponding tropospheric  $O_3$  column increases slightly. The change of  $CH_4$  lifetime attributed to shipping emissions averaged over the analysed 10 years is large compared to the corresponding  $NO_x$  emission increase of 0.75 Tg(N) between CCMI-1 and CCMI-2022. The difference of  $NO_x$  emissions from shipping between CCMI-1 and CCMI-2022 has a distinct temporal evolution peaking in the year 2008 (see Fig. S8), when changes of tropospheric  $O_3$  and  $CH_4$  lifetime of this category are most pronounced. In contrast to the other prescribed emission categories, emissions of  $NO_x$  from aviation are reduced in CCMI-2022 compared to CCMI-1, which is reflected by a decrease of the tropospheric  $O_3$  column and an extension of  $CH_4$  lifetime. In addition, the artificially shifted

Table 3. Overview of difference of the prescribed emissions, the tropospheric  $O_3$  column, and the tropospheric  $CH_4$  lifetime for the individual tagging categories between the simulations EMIS-02 and EMIS-01. For the contribution of tropospheric  $CH_4$  lifetime change, the estimates derived using method 2 are shown (see Sect. 2.4).

| Category                  | $\Delta \mathrm{NO_x}$ |       | $\Delta 	ext{CO}$ |       | $\Delta$ NMHC |       | $\Delta O_3$ | $\Delta \mathrm{CH_4}$ lifetime |
|---------------------------|------------------------|-------|-------------------|-------|---------------|-------|--------------|---------------------------------|
|                           | [Tg(N)]                | [%]   | [Tg(C)]           | [%]   | [Tg(C)]       | [%]   | [DU]         | [years]                         |
| Northern Hemisphere       |                        |       |                   |       |               |       |              |                                 |
| anthropogenic non-traffic | 3.08                   | 20.1  | 7.25              | 4.5   | 13.70         | 21.0  | 0.76         | -0.15                           |
| land transport            | 2.99                   | 34.5  | 6.68              | 10.5  | 6.44          | 42.0  | 0.64         | -0.10                           |
| biomass burning           | -0.21                  | -7.1  | -36.35            | -38.6 | 10.00         | 103.6 | 0.07         | -0.08                           |
| shipping                  | 0.75                   | 14.5  | -0.23             | -48.4 | 0.19          | 9.8   | 0.16         | -0.08                           |
| N2O                       | -                      |       | -                 |       | -             | -     |              | 0.0                             |
| lightning                 | 0.0                    | *     | -                 |       | -             |       | 0.03         | 0.01                            |
| stratosphere -            |                        |       | -                 |       | -             |       | 0.07         | 0.01                            |
| CH4                       | _                      |       | -                 |       | -             |       | 0.00         | 0.02                            |
| aviation                  | -0.09                  | -10.2 | 0.21              | _     | 0.07          | _     | -0.07        | 0.06                            |
| biogenic                  | 0.0*                   |       | 0.0               | 0.0   | 0.0*          |       | -0.27        | 0.07                            |
| residuum                  | -                      |       | -                 |       | -             |       | -            | -0.01                           |
| total                     | 6.51                   | 16.5  | -22.44            | -6.6  | 30.41         | 9.8   | 1.39         | -0.25                           |
| Southern Hemisphere       |                        |       |                   |       |               |       |              |                                 |
| anthropogenic non-traffic | 0.34                   | 21.7  | 1.24              | 5.3   | 8.67          | 101.2 | 0.51         | -0.13                           |
| land transport            | 0.03                   | 2.6   | 1.06              | 13.3  | 0.91          | 32.4  | 0.02         | 0.03                            |
| biomass burning           | 1.62                   | 64.6  | -8.54             | -10.4 | 16.88         | 196.0 | 0.36         | -0.19                           |
| shipping                  | 0.53                   | 76.4  | -0.01             | -20.7 | 0.03          | 12.6  | 0.37         | -0.20                           |
| N2O                       |                        |       | -                 |       | -             |       | 0.02         | 0.0                             |
| lightning                 | lightning 0.0*         |       | -                 |       | -             |       | 0.02         | -0.02                           |
| stratosphere -            |                        | -     |                   | -     |               | 0.03  | 0.0          |                                 |
| CH4                       | -                      |       | -                 |       | -             |       | 0.09         | 0.02                            |
| aviation                  | -0.02                  | -28.8 | 0.01              | _     | 0.005         | _     | -0.06        | 0.04                            |
| biogenic 0.               |                        | *     | 0.0               | 0.0   | 0.0           | )*    | -0.18        | 0.09                            |
| residuum                  |                        |       |                   |       | -             |       | -            | -0.05                           |
| total                     | 2.51                   | 25.4  | -6.24             | -4.5  | 26.49         | 12.1  | 1.19         | -0.4                            |

<sup>\*</sup> Identical online calculated emissions as assured by simulation set-up (see Sect. 2.1).

 $<sup>^{\</sup>diamond}$  No CO and NMHC emissions from sector aviation in simulation EMIS-01.

Figure 8. Change of tropospheric  $CH_4$  lifetime attributed to individual tagging categories, separately for (a) the Northern Hemisphere and (b) the Southern Hemisphere. The black bars show the  $CH_4$  lifetime with respect to the oxidation with total OH,  $\tau_{CH_4,OH_{total}}$ , for the simulations EMIS-01 (CCMI-1) and EMIS-02 (CCMI-2022). The blue and red bars show the contribution of category i to the  $CH_4$  lifetime change with respect to total OH,  $\Delta\tau_{CH_4,OH_i}$ , calculated either using method 1 (blue) or method 2 (red, see Sect. 2.4 for details on the derivation). The rightmost blue and red bars represent the sum of  $\Delta\tau_{CH_4,OH_i}$  over all categories.

aviation emissions from the tropics to the polar regions in the CEDS emission inventory, as reported by Thor et al. (2023), can play a role here.

In the SH, the anthropogenic non-traffic, biomass burning, and shipping categories contribute most to the changes of the tropospheric  $O_3$  column and the  $CH_4$  lifetime. In these sectors, the increase (CCMI-2022 vs CCMI-1) of  $NO_x$  emissions is most pronounced. In addition, CO emissions from biomass burning are also reduced (CCMI-2022 compared to CCMI-1) in the SH. In the SH, the tropospheric  $O_3$  column and the  $CH_4$  lifetime are more sensitive to emission changes, e.g. anthropogenic non-traffic  $NO_x$  emission changes are 3.08 Tg(N) in the NH and 0.34 Tg(N) in the SH, which corresponds to increases of the tropospheric  $O_3$  column of 0.76 DU and 0.51 DU, respectively. The land transport and aviation categories play a minor

role in the SH. Accordingly, these categories hardly contribute to the tropospheric  $O_3$  column and the  $CH_4$  lifetime difference between the two simulations.

The biogenic category counteracts the increase of the tropospheric  $O_3$  column and the shortening of the  $CH_4$  lifetime in both hemispheres, although biogenic  $NO_x$ , CO and NMHC emissions are unchanged (see Tab. 3 and Sect. 2.1). The  $O_3$  production efficiency of one category can be affected by emission changes of another category due to non-linear compensation effects (see e.g., Mertens et al., 2018, their Fig. 6). In this case, the diagnosed contribution of the biogenic category to changes of  $O_3$  and  $CH_4$  lifetime is not caused by emission changes, but by a larger  $O_3$  burden efficiency (see Fig. 4).

# 420 4 Discussion



#### 4.1 Limitations

First of all, the used TAGGING method relies on some simplifications, which are discussed in more detail by Grewe et al. (2017); Rieger et al. (2018); Mertens et al. (2018). As example, decomposition of PAN from lightning can lead to artificial small contributions of NMHCs from lightning emissions. For the TAGGING of  $HO_x$ , a reduced set of chemical reactions is considered, compared to the chemical mechanism calculated by MECCA (Rieger et al., 2018). This means that the sum of all tagged OH tracers can deviate from the total OH tracer. The  $CH_4$  loss unaccounted by the TAGGING, i.e. represented by the residuum lifetime  $\tau_{CH_4,OH_{res}}$  (see Eq. 5), is, however, less important than any of the tagged categories in our simulations. We calculate the contribution of the unaccounted loss to the  $CH_4$  lifetime change with respect to total OH, which is -0.01 years in the NH and -0.05 years in the SH (see Tab. 3). Categories with a smaller contribution are not considered to have a meaningful effect on the  $CH_4$  lifetime change, i.e.  $N_2O$ , lightning and stratosphere in the NH, and land transport,  $N_2O$ , lightning, stratosphere,  $CH_4$  and aviation in the SH. Further, we calculate the contribution to the  $CH_4$  lifetime change using two methods, which are explained in Sect. 2.4. The relative contributions are consistent for both methods, but only method 2 is closed, meaning that the sum of all individual contributions results in the total change of  $CH_4$  lifetime. Therefore, we recommend to use method 2 for further studies.

Further, our simulation setup does not allow to separate the effects of the individual species, CO, CO,

# **4.2** Comparison to literature




As we show in the results section, the updated prescribed emission inventory leads to an increase of the tropospheric  $O_3$  column by about 4%. As tropospheric  $O_3$  is generally overestimated by the EMAC model (Jöckel et al., 2016), the use of the CCMI-2022 emission inventory for  $O_3$  precursor species brings model's tropospheric  $O_3$  further away from observations. Similarly, the tropospheric  $CH_4$  lifetime is generally underestimated by EMAC compared to observations (Jöckel et al., 2016), which means that the use of the CCMI-2022 emission inventory leads to a less good agreement with observations. This is a common issue of many models (see also Prather and Zhu, 2024).

Previous studies analysing simulation results from other chemistry-climate models prescribing the CEDS emission inventory for O<sub>3</sub> precursor emissions noted a strong shortening of the CH<sub>4</sub> lifetime after the year 1990 as well. For instance, Skeie et al. (2023, their Figure 3) show that the increase of OH is steeper in the AerChemMIP multi-model mean ensemble, which is based on the CEDS emission inventory, compared to the multi-model mean ensemble of CCMI-1. Similarly, the three chemistry-climate models of the AerChemMIP ensemble as analysed by Stevenson et al. (2020) show a drop of the CH<sub>4</sub> lifetime after the year 1990 in the historical simulation, which is not apparent when O<sub>3</sub> precursor emissions are kept at pre-industrial levels. Thus, these studies indicate that the CEDS emission inventory leads to similar effects also in other CCMs, but did not aim at attributing differences to individual emission sectors.

It is noteworthy that both simulations show negative temporal trends of the  $CH_4$  lifetime and positive trends of OH over the period 2000 - 2010, which is in line with the trends simulated by other CCMs (Nicely et al., 2020; Zhao et al., 2020; Stevenson et al., 2020). In line with this, the study by Morgenstern et al. (2025) indicates a positive trend of OH since 1997 based on SH measurements of  $^{14}CO$ . On the contrary, previous observational based estimates and inversions of OH do not indicate a trend (see IPCC, 2021, (Table 6.7) and references therein).

The CEDS<sub>GBD-MAPS</sub> emission inventory provided by McDuffie et al. (2020) is an update of CEDS (Hoesly et al., 2018), which suggests smaller global emissions of NO<sub>x</sub> and CO after the year 2006 compared to CEDS. The difference is about 3 Tg(N) for NO<sub>x</sub> and 10 Tg(C) for CO in the year 2010 (McDuffie et al., 2020, their Figure 6). The differences of NO<sub>x</sub> emissions are explained by reduced emissions from the transport sector in India and Africa, and by an updated emission estimate for China (Zheng et al., 2018). The reduction of NO<sub>x</sub> emissions in CEDS<sub>GBD-MAPS</sub> would compensate about 34% of the NO<sub>x</sub> emissions increase from EMIS-01 to EMIS-02, and the increase of CO emissions would compensate about 33% of the reduction of CO emissions from EMIS-01 to EMIS-02. For NMHCs, McDuffie et al. (2020) suggest 5% larger global emissions than Hoesly et al. (2018), which result mainly from larger emissions in Africa from the oil and gas sector.

# 470 5 Conclusions

In this study, we analyse the impact of the two different emission inventories for  $NO_x$ , CO, and NMHCs, as have been used for the CCMI-1 and CCMI-2022 initiatives, on the simulated tropospheric  $O_3$ , OH, and  $CH_4$  lifetime. Therefore, we have performed targeted simulations with the EMAC model over the years 2000 - 2010 equipped with the diagnostic TAGGING

submodel to further attribute differences of  $O_3$ , OH, and  $CH_4$  lifetime to individual emission sectors (Grewe et al., 2017; Rieger et al., 2018).

Our results suggests that the use of the emission inventory of CCMI-2022 enhances the tropospheric  $O_3$  column by 3.7% in the NH, and by 4.5% in the SH in comparison to the emission inventory used for CCMI-1. The tropospheric  $CH_4$  lifetime with respect to the oxidation with OH shortens by 0.25 years in the NH, and by 0.4 years in the SH.

Using the TAGGING submodel, we attribute these differences to individual emission sectors. In the NH, the sectors contributing most to the change in the tropospheric  $O_3$  column are anthropogenic non-traffic (55%), land transport (46%), shipping (12%), and biomass burning (5%). In the SH, the primary contributors are anthropogenic non-traffic (43%), biomass burning (30%), and shipping (31%). The aviation sector counteracts the increase of the tropospheric  $O_3$  column by about 5% in both hemispheres. Due to the non-linearity of the  $O_3$  chemistry, the contributions of tagging categories with unmodified emissions also change. For instance, the  $O_3$  production efficiency of biogenic soil  $NO_x$  emissions decreases, which counteracts the increase of the tropospheric  $O_3$  column by 19% in the NH, and by 15% in the SH.







Our study is to our knowledge the first that uses the tagging approach to attribute the difference of  $CH_4$  lifetime between two simulations to individual emission sectors. We find that the sectors, which contribute most to the increase of the tropospheric  $O_3$  column, also contribute most to the shortening of the tropospheric  $CH_4$  lifetime. However, the contribution of one sector to either the change of  $O_3$  or  $CH_4$  lifetime is not necessarily the same. For instance, in the NH the biomass burning sector contributes about 12% to the increase of the tropospheric  $O_3$  column, but about 32% to the shortening of the  $CH_4$  lifetime.

Further, our results demonstrate the dependence of tropospheric  $O_3$  and the  $CH_4$  lifetime simulated by CCMs on the prescribed emission inventories for  $NO_x$ , CO, and NMHCs and the resulting uncertainty. The tropospheric  $O_3$  column and the  $CH_4$  lifetime differ by about 4% for the two used emission inventories that both aim at representing recent historical conditions. It would be valuable to investigate whether the change of  $O_3$  and  $CH_4$  lifetime caused by the modified emission inventory are of similar magnitude in other CCMs. The studies by Skeie et al. (2023) and Stevenson et al. (2020) suggest that the CEDS emission inventory leads to a significant shortening of the  $CH_4$  lifetime after the year 1990 in other CCMs as well.

Future research utilizing the tagging method could provide deeper insights into this topic. For example, additional simulations with a revised setup, where only one chemical species (e.g.,  $NO_x$ ) is modified within the emission inventories, would enable a more straightforward comparison of the contributions of  $NO_x$ , CO, or NMHC emissions to  $O_3$ , OH and the  $CH_4$  lifetime in the troposphere. Regional tagging would be a valuable addition, enabling closer examination of regional responses.

Code and data availability. The Modular Earth Submodel System (MESSy; doi: 10.5281/zenodo.8360186) is continuously further developed and applied by a consortium of institutions. The usage of MESSy and access to the source code is licenced to all affiliates of institutions which are members of the MESSy Consortium. Institutions can become a member of the MESSy Consortium by signing the MESSy Memorandum of Understanding. More information can be found on the MESSy Consortium Website (http://www.messy-interface.org). The simulation results presented here are based on MESSy version 2.55.2. The simulation results are archived at zenodo and available under the DOIs 10.5281/zenodo.14712802 (EMIS-01 simulation) and 10.5281/zenodo.14712940 (EMIS-02 simulation).

# Appendix A: Derivation of methane lifetime change formula

In this Appendix, we explain how we derived Eq. 6 to calculate  $\Delta \tau_{CH_4,OH_i}$ , the contribution of category i to the CH<sub>4</sub> lifetime change with respect to total OH. The reciprocal of the CH<sub>4</sub> lifetime with respect to total OH is given by the sum of the reciprocals of the individual lifetimes (see Eq. 4). Here, we use  $\tau_{CH_4,OH_{sum}}^{\star}$ , so that the CH<sub>4</sub> loss by the total OH including the residuum is accounted for.

$$\frac{1}{\tau_{CH_4,OH_{sum}}^{\star}} = \sum_{j} \frac{1}{\tau_{CH_4,OH_j}}$$

$$= \frac{1}{\tau_{CH_4,OH_i}} + \sum_{j \neq i} \frac{1}{\tau_{CH_4,OH_j}}$$
(A1)

Thus, we can write  $\tau^{\star}_{CH_4,OH_{sum}}$  as a function of  $\tau_{CH_4,OH_i}$ 

$$\tau_{CH_4,OH_{sum}}^{\star}(\tau_{CH_4,OH_i}) = \left(\frac{1}{\tau_{CH_4,OH_i}} + \sum_{j \neq i} \frac{1}{\tau_{CH_4,OH_j}}\right)^{-1},\tag{A2}$$

for which the derivative with respect to  $\tau_{CH_4,OH_i}$  is given by

$$\frac{d\tau_{CH_4,OH_{sum}}^{\star}}{d\tau_{CH_4,OH_i}} = \left(\frac{1}{\tau_{CH_4,OH_i}} + \sum_{j \neq i} \frac{1}{\tau_{CH_4,OH_j}}\right)^{-2} \cdot \tau_{CH_4,OH_i}^{-2}$$

$$= \frac{\left(\tau_{CH_4,OH_{sum}}^{\star}\right)^2}{\left(\tau_{CH_4,OH_i}\right)^2} \tag{A3}$$

The derivative  $\frac{d\tau_{CH_4,OH_{sum}}^c}{d\tau_{CH_4,OH_i}}$  indicates by how much the  $\mathrm{CH}_4$  lifetime with respect to total OH changes with changing  $\tau_{CH_4,OH_i}$ . Thus, we approximate the contribution of category i to the  $\mathrm{CH}_4$  lifetime change with respect to total OH,  $\Delta\tau_{CH_4,OH_i}$ , with this derivative

$$\Delta \tau_{CH_4,OH_i} = \frac{(\tau_{CH_4,OH_{sum}}^{\star})^2}{(\tau_{CH_4,OH_i})^2} \cdot (\tau_{CH_4,OH_i}(\text{EMIS-02}) - \tau_{CH_4,OH_i}(\text{EMIS-01})).$$
 (A4)

*Author contributions*. CA analysed the data, created most of the plots and drafted the manuscript. LS, MM, PJ developed the concept of the study and prepared the simulation setup. All authors contributed to the interpretation of the model results and to writing the manuscript.

Competing interests. At least one author is member of the editorial board of ACP.

Disclaimer: TEXT

Acknowledgements. We thank Alina Fiehn (DLR) for doing the internal review. We used Climate Data Operators (CDO; https://code.mpimet.mpg.de/projects/cdo/, last access: 13 November 2024; Schulzweida, 2023) and netCDF Operators (NCO; Zender, 2024) for data processing. Further, we used Python, especially the packages Xarray (Hoyer and Hamman, 2017) and Matplotlib (Hunter, 2007), for data analysis and producing the figures. We furthermore thank all contributors of the project ESCiMo (Earth System Chemistry integrated Modelling), which provides the model configuration and initial conditions for the model simulations performed here. This work used resources of the Deutsches
 Klimarechenzentrum (DKRZ) granted by its Scientific Steering Committee (WLA) under project id0853. Further, datasets provided by MESSy via the DKRZ data pool were used.

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
