# Peer review of "Effects of different emission inventories on tropospheric ozone and methane lifetime"

_EGUsphere, 2025_

## Author Comment (AC1)

**Reply to reviewer #2**

Catherine Acquah[1], Laura Stecher[1,a], Mariano Mertens[1,2], and Patrick Jöckel[1]

[1]Deutsches Zentrum für Luft- und Raumfahrt, Institut für Physik der Atmosphäre, Oberpfaffenhofen, Germany
[2]Faculty of Aerospace Engineering, Section Operations and Environment, Delft University of Technology, 2629 HS, Delft, The Netherlands
[a]now at: Yusuf Hamied Department of Chemistry, University of Cambridge, Cambridge, CB2 1EW, United Kingdom

We thank reviewer #2 for her/his comments and the evaluation of our paper. Below, we repeat each comment (in blue) and address it (in black). Changes of the manuscript are written in italics.

The manuscript "Effects of different emission inventories on tropospheric ozone and methane lifetime" by Acquah et al. presents a modeling study of the effects of differing emissions inventories on metrics of climate and air quality interest, namely, tropospheric ozone and hydroxyl radical (OH) burdens and the resulting methane lifetime. Based on two simulations performed by the EMAC model for two different phases of the Chemistry Climate Model Initiative (CCMI), each with its own prescribed emissions inventories, the authors observed significant differences in the metrics listed above and go on to evaluate the sectors (industrial, land transport, shipping, etc.) driving those differences using a tagging approach. Tropospheric ozone is more abundant in the simulation using more up-to-date emissions and methane lifetime is shorter in both hemispheres. Some budget closure calculations are also performed to understand the sectoral contributions for methane lifetime, and burden efficiency calculations are performed to understand the influence of nitrogen oxides (NOx) emissions changes specifically on ozone burden in a given sector.

Overall I find the presented analysis to be reasonably thorough for a model-focused study, well polished, and clear in its conclusions and methods. The findings are not "ground-breaking" but are instead, in my mind, an important contribution by quantifying and systematically seeking to understand a phenomenon that modeling groups often know is happening but is not always well characterized. By describing in detail the changes in tropospheric ozone, OH, and methane lifetime that might be expected from a routine update or switch of emissions inventories, the authors are assisting the modeling community and those seeking to constrain present-day and historical values of these quantities across models and through time. The authors are clear about the limitations of the study and also do a good job of citing past work and putting into context some of the specific changes in emissions sectors and species. I offer some minor comments for clarification, readability, and correction, but otherwise regard this manuscript as a strong candidate for publication in ACP.

Thank you very much for this positive feedback!

**1 Minor comments**

L37: In this paragraph, sources of precursors are described but seem to me incomplete – first there's anthropogenic NOx, then natural NOx. Then anthropogenic CO only, and natural NMHC only. Why not also include natural CO (biomass burning, to the degree it is natural, and from natural methane, I presume?), and anthropogenic NMHC (industry)?

We added the following information in the revised manuscript:

line 36 (of the preprint): *Natural emissions of* CO *occur during the combustion of biomass, e.g. lightning induced fires (Zheng and Zhao, 2019), and from the oxidation of hydrocarbons, but also from plants and the oceans (Khalil and Rasmussen, 1990).*

line 37 (of the preprint): *The main anthropogenic sources of NMHC include the incomplete combustion of fossil fuels, petroleum from geological reservoirs, and the distillation and distribution of oil and gas products (Pozzer et al., 2010).*

L47: For this sentence, if the authors deem a citation helpful, Duncan et al., 2024 might be an appropriate reference, as it discusses the infeasibility of global observations of OH:

Duncan, B. N., et al.: Opinion: Beyond global means – novel space-based approaches to indirectly constrain the concentrations of and trends and variations in the tropospheric hydroxyl radical (OH), Atmos. Chem. Phys., 24, 13001–13023, https://doi.org/10.5194/acp-24-13001-2024, 2024.

Thank you, we added the citation as suggested.

L122: I'm not familiar with this phrase, "binary identical". I don't see how meteorology can be binary, but are met-dependent emissions binary, as in, either on or off? Please clarify or remove "binary".

We think that there is a misunderstanding in the understanding of "binary identical" here. What we want to express is that due to the use of the QCTM mode (see manuscript), differences between the emission inventories that lead to differences in the chemical composition do not affect the meteorology, not even numerically. Therefore, the simulated meteorological quantities are "binary identical" in all simulations. In other words there are no numerical differences between both simulations, and therefore the online calculated emissions that depend on the meteorology are exactly the same in both simulations.

Previous version in preprint:

l. 116 (of preprint): *This assures that differences of the emission inventories prescribed in the two simulations do not affect the model's simulated meteorology.*

l. 122 (of preprint): *Due to the binary identical meteorology between the simulations, these online calculated, meteorology dependent emissions are also binary identical in both simulations.*

Updated to avoid binary identical:

l. 116 (of preprint): *This assures that differences of the emission inventories prescribed in the two simulations do not affect the model's simulated meteorology, not even numerically.*

l. 122 (of preprint): *Due to the identically simulated meteorology in all simulations due to the QCTM mode, these online calcu-*

*lated, meteorology dependent emissions are exactly the same in both simulations with no differences, not even numerical noise.*

L150: For Figs. S6, S7, and S8, it would be helpful to include in legend that solid lines represent EMIS-01, dashed lines represent EMIS-02 instead of it being buried in one of the captions.

We have updated Figures S6, S7 and S8 in the supplement accordingly.

**2 Technical corrections**

– L28: Not sure that citing Seinfeld and Pandis twice in this sentence is necessary; once at the end would convey the same attribution, in my mind. Removed in the middle of the sentence.

– L60: Should this state "CCMI-2022" rather than "CCMI-2"? Corrected.

– L74: "targeted" misspelled Corrected.

– L150: In Fig. S8, "anthropogenic" is misspelled in legend. Corrected.

– L151: 1010 should be 2010 Corrected

– L261: Remove "hemisphere," redundant Corrected

– L271: Fig. S12 y-axis label has misspelling of "efficiency" Corrected.

– L286: Fig. 4 y-axis label "efficiency" misspelled Corrected.

– L334: One of these EMIS-01's should be an EMIS-02 I presume; directionality dependent on whether "CH4," as stated, or "CH4 lifetime" is the quantity being compared. We have corrected it, and we meant "CH4 lifetime".

– L413: Both instances of "from EMIS-01 to EMIS-01" in this sentence should instead state "from EMIS-01 to EMIS-02" I believe Yes, you are right. Corrected.

– L433: remove one "that" Corrected.

– L452: "available" misspelled Corrected.

**References**

Khalil, M. and Rasmussen, R.: The global cycle of carbon monoxide: Trends and mass balance, Chemosphere, 20, 227–242, https://doi.org/10.1016/0045-6535(90)90098-E, 1990.

Pozzer, A., Pollmann, J., Taraborrelli, D., Jöckel, P., Helmig, D., Tans, P., Hueber, J., and Lelieveld, J.: Observed and simulated global distribution and budget of atmospheric $C_2$-$C_5$ alkanes, Atmospheric Chemistry and Physics, 10, 4403–4422, https://doi.org/10.5194/acp-10-4403-2010, 2010.

Zheng, B., C. F. Y. Y. C. P. F.-C. A. D. M. N. P. R. J. W. Y. W. H. M. and Zhao, Y.: Global atmospheric carbon monoxide budget 2000– 2017 inferred from multi-species atmospheric inversions, Earth Syst. Sci. Data, 11, 1411–1436, https://doi.org/https://doi.org/10.5194/essd-11-1411-2019, 2019.

85

---

## Author Comment (AC2)

**Reply to reviewer #1**

Catherine Acquah[1], Laura Stecher[1,a], Mariano Mertens[1,2], and Patrick Jöckel[1]

[1]Deutsches Zentrum für Luft- und Raumfahrt, Institut für Physik der Atmosphäre, Oberpfaffenhofen, Germany
[2]Faculty of Aerospace Engineering, Section Operations and Environment, Delft University of Technology, 2629 HS, Delft, The Netherlands
[a]now at: Yusuf Hamied Department of Chemistry, University of Cambridge, Cambridge, CB2 1EW, United Kingdom

We thank reviewer #1 for her/his comments and the evaluation of our paper. Below, we repeat each comment (in blue) and address it (in black). Changes of text in the manuscript are written in italics.

The manuscript "Effects of different emission inventories on tropospheric ozone and methane lifetime" investigates the in-
5  fluence of ozone precursor emissions on model simulations of ozone concentrations and methane lifetime. The authors applied a tagging approach to attribute differences in these variables to specific emission sectors. The manuscript is well-written, and the results provide valuable insights into inter-model differences in tropospheric ozone and OH concentrations. I recommend the manuscript for publication after addressing the following minor comments:

Thank you very much for this positive evaluation!

10

L120: Are there other non-methane volatile organic compounds included in the biogenic emissions besides C5H8

For the other NMHCs of biogenic origin a prescribed climatology from GEIA is used, which is the same in all simulations. We added the following information:

*In addition to the online calculated natural emissions, a climatology of biogenic emissions of NMHCs and* $CO$ *is prescribed*
15  *from the Global Emissions InitiAtive (GEIA) in all simulations.*

Line 150 and Table 3: It would be beneficial to include both absolute and relative differences in emissions between the EMIS-01 and EMIS-02 simulations. This would provide a clearer comparison of the two emission inventories.

We have included the relative differences in the Table of the revised manuscript.

20

Section 2.2.3 The TAGGING submodel is used to attribute O3 production and OH mixing ratios to different emission sectors. However, given the highly nonlinear chemistry of O3 and OH, a more detailed explanation of how the TAGGING method attributes O3 and OH to individual emissions sectors would enhance the reader's understanding of the results.

We are hesitating to provide a lot more details here, since they can hardly be complete, without duplicating the entire paper by
25  Grewe et al. (2017). Nevertheless, we reformulated large parts of the section, now also explicitly stating that the non-linearity of the ozone chemistry is taken into account by the tagging method.

In equation (9), are there other ozone precursors (e.g. CO, NMVOC) also contributing to B(O3)? Are the differences in burden efficiency influenced by other ozone precursors?

30  Indeed, the burden efficiency is calculated only with respect to $NO_x$ emissions. Especially the biogenic sector has a lot of VOC emissions, but lower $NO_x$ emissions (due to soil-$NO_x$) emissions. The VOC emissions also contribute to ozone formation, which can lead to a larger burden efficiency with respect to $NO_x$ emissions compared to anthropogenic sectors with rather large $NO_x$, but lower VOC emissions. Since we only use this for comparing the results between the two simulations (but not for assessing the role of individual sectors), we prefer to not further detail this in the text.

35

The authors use two different methods to calculate changes in CH4 lifetime attributable to individual emission sources. While the two methods yield similar results for most sectors, they show divergent results for the land transportation sector (in Figure 7, method 1 indicates a large negative contribution, while method 2 shows a positive contribution). This discrepancy warrants further explanation. I recommend that the authors calculate the global tropospheric CH4 reaction-weighted OH con-
40  centrations contributed by each emission sector and simulation. This would provide a clearer understanding of how emissions influence CH4 lifetime.

We are not sure what is meant by "(in Figure 7, method 1 indicates a large negative contribution, while method 2 shows a positive contribution". Could it be that the two methods (indicated by red and blue bars) and the separation into SH and NH (shown in panel a and b) have been confused by the reviewer? Figure 7 shows that the contribution of the land traffic sector
45  to the $CH_4$ lifetime change estimated by **both methods is negative in the NH**, whereas **both methods indicate a positive contribution in the SH**.

We followed the suggestion and added a barplot showing the tropospheric OH of individual tagging categories weighted by the reaction with $CH_4$ (following Lawrence et al. (2001)) to the revised manuscript. The plot shows the global mean OH, as well as OH in both hemispheres separately.

50

L354 "the CH4 reduction from EMIS-01 to EMIS-01 in the NH". I think it should be "the CH4 lifetime reduction from EMIS-01 to EMIS-02 in the NH".

You are right. Thank you for pointing this out. We corrected it in the revised manuscript.

**References**

55   Grewe, V., Tsati, E., Mertens, M., Fromming, C., and Jockel, P.: Contribution of emissions to concentrations: the TAGGING 1.0 submodel based on the Modular Earth Submodel System (MESSy 2.52), Geoscientific Model Development, 10, 2615–2633, https://doi.org/10.5194/gmd-10-2615-2017, 2017.

Lawrence, M. G., Jöckel, and von Kuhlmann, R.: What does the global mean OH concentration tell us?, Atmos. Chem. Phys., 1, 37–49, https://doi.org/10.5194/acp-1-37-2001, 2001.

---

## Author Comment (AC3)

**Reply to reviewer #3**

Catherine Acquah[1], Laura Stecher[1,a], Mariano Mertens[1,2], and Patrick Jöckel[1]

[1]Deutsches Zentrum für Luft- und Raumfahrt, Institut für Physik der Atmosphäre, Oberpfaffenhofen, Germany
[2]Faculty of Aerospace Engineering, Section Operations and Environment, Delft University of Technology, 2629 HS, Delft, The Netherlands
[a]now at: Yusuf Hamied Department of Chemistry, University of Cambridge, Cambridge, CB2 1EW, United Kingdom

We thank reviewer #3 for her/his comments and the evaluation of our paper. Below, we repeat each comment (in blue) and address it (in black). Changes of the manuscript are written in italics.

This is a very nice paper that examines the response of some key climate variables to model input data from two versions of emissions from the Chemistry-Climate Model Intercomparison CCMI project. The authors use a well-established chemistry-climate model together with some innovative process-level outputs to investigate the response of tropospheric ozone, hydroxyl radical and methane lifetime to changes in emissions between two eras of CCMI. This work is significant for understanding both the sensitivity of models to evaluated emissions data and to begin to describe the source of intermodel diversity by defining model sensitivities. The study is well-designed and this paper fits well within the remit of ACP.

Thank you very much for this positive feedback!

After a short introduction, Section1, Section 2 describes the study's Methods and Data. Section 2.1 describes the model in very short detail, and could be combined with section 2.2.1 which currently describes the components and other options chosen, as well as some of the emissions and with Section 2.2.2 which describes other emissions. Overall, I think the MS would be better if all the chemical species - methane, N2O, NOx and CO, BVOC etc- were discussed together in one section, and if L114-119 and L124-126 were moved up the MS. Similarly I think all of the text from L1-L20 of the Supplementary should be moved either to this section, or the Discussion.

We modified the structure of the methods section. It now starts with a general short description of MESSy, followed by set-up specifics (i.e. nudging, QCTM mode, and natural emissions), and ends with a description of the different simulations that were performed for this paper. We still prefer to introduce the prescribed inventories of ozone precursor species in a separate section as they are described quite extensively.

Concerning the suggestion to move L1-L20 of the Supplement to the main manuscript, we are hesitating as in the supplement changes in the simulations originally performed for the two phases of CCMI are discussed. These differences do not affect the simulations analysed in the manuscript, EMIS-01 and EMIS-02, which have been run with the same model version. We modified the supplement to stress this point.

Section 2.2.3 describes the tagging methodology, and 2.3 the calculation of methane lifetime. In this section, and in the MS overall, I think the word category becomes rather laborious. Where the categories are emissions from a specific sector, it would be helpful, and the MS would be improved, if the word 'sector' was retained in favour of 'category', and the other non-emission tagging labels were called e.g. 'N2O decomposition processes' or something similar. (The authors in fact seem to prefer the word sector in their Conclusions section).

We are hesitating to avoid the term category completely, as it is the term used by the publications on the implementation of the TAGGING submodel to describe all the different sources regardless of the type of source (emission sector, decomposition process) (Grewe et al., 2017; Rieger et al., 2018). Therefore, we think "category" is the right term to use when describing technical details of the TAGGING submodel or the methods to attribute the methane lifetime change. The description of the methodology should be independent of the type of source, e.g. emission, decomposition process, production in the stratosphere. Later on in the results section, we want to attribute O3, OH and CH4 lifetime changes to individual emission sectors as the prescribed emissions are different in both simulations. We added a short introduction/definition of both terms, (tagging) category and (emission) sector, in the methods section when introducing the TAGGING submodel to avoid confusion about the two terms.

*The term category is used independently of the type of source, e.g. emission, decomposition process, or production in the stratosphere. In contrast, we use the term (emission) sector only for emissions. For example, the category land transport represents the contribution of the land transport emission sector, whereas the category stratosphere represents downward transport of $O_3$ formed in the stratosphere, which is not related to any emission sector.*

In section 2.3, on methane lifetime, the terminology used there might also be confusing to some readers - 'total' lifetime is often defined to include other loss processes, e.g. stratospheric loss, loss with Cl - so would it be better to define terms and specifying OH consistently and explicitly, e.g. L220 the term 'total lifetime', tau_CH4, OH_total, would be better as 'total OH lifetime'. Similarly for L201 tau_CH4,sum would appear to be tau_CH4,sum_OH.

You are right that this could be confusing. We revised Sect. 2.3. and the Appendix and hope that it is clear now that the CH4 lifetime with respect to total OH is meant.

Accordingly, we also changed the acronym names from $\tau_{CH_4,sum}$ to $\tau_{CH_4,OH_{sum}}$, from $\tau_{CH_4,res}$ to $\tau_{CH_4,OH_{res}}$, and from $\Delta\tau_{CH_4,i}$ to $\Delta\tau_{CH_4,OH_i}$.

Section 3 presents the results. §3.1 highlights the change in the tropospheric ozone column between the two experiments. It would be interesting to know whether these changes bring the model into better or worse agreement with observational data on tropospheric ozone columns, if available.

Unfortunately, the update of the emission inventory from CCMI-1 to CCMI-2022 does not bring the model results into a better agreement with observed tropospheric ozone columns. The tropospheric ozone bias (and the corresponding underestimated tropospheric methane lifetime) become even larger (shorter). We refer to Jöckel et al. (2016) for a detailed analysis (CCMI-1)

and point out that this tropospheric ozone / methane lifetime bias is a common problem of many models (see also Prather and Zhu, 2024).

We added to the discussion section of the revised manuscript:

*As we show in the results section, the updated prescribed emission inventory leads to an increase of the tropospheric $O_3$ column by about 4%. As tropospheric $O_3$ is generally overestimated by the EMAC model (Jöckel et al., 2016), the use of the CCMI-2022 emission inventory for $O_3$ precursor species brings model's tropospheric $O_3$ further away from observations. Similarly, the tropospheric $CH_4$ lifetime is generally underestimated by EMAC compared to observations (Jöckel et al., 2016), which means that the use of the CCMI-2022 emission inventory leads to a less good agreement with observations. This is a common issue of many models (see also Prather and Zhu, 2024).*

Consistent use of the labels from Table 2 and in this section would help ('N2O' category vs 'N2O decomposition' category). Corrected.

The discussion of the drivers of the ozone column changes would be improved by diagnosing the response of O3P and O3L budget terms, if available for these tagged experiments. Including a calculation of ozone production efficiency would allow comparisons with other studies.

We added a figure showing zonal mean effective production and loss of $O_3$ in the supplement and describe the differences between the two simulations in the main manuscript shortly.

Indeed, we initially calculated the ozone production efficiency (OPE), but we decided at the end to not show and discuss it in the present study. We took this decision is for several reasons: First, OPE is rather a qualitative metric only and sometimes difficult to interpret. Second, the analyses we performed were not very conclusive, the regional distribution of OPE did hardly change between the different simulations, most probably since the spatial (horizontal) distributions of emissions between the CCMI-1 and CCMI-2022 inventories are very similar, but only the magnitudes of emission fluxes are different. And third, we want to keep the focus of your manuscript on the attribution of methane lifetime to different emission sectors.

Additionally, supplementary plots of zonal mean emission changes (latitude vs altitude) could help explore how emission changes impact ozone distribution.

We think that such altitude-latitude plots are not really meaningful, since most sectors are emitted in the lowermost layers/close to the surface, except for aviation, which shows a very simlar methane lifetime contribution between CCMI-1 and CCMI-2022 (see Figure 7 in the manuscript), and lightning NOx, which does not change between the simulations.

3.2 discusses OH and shows the absolute-scale change in OH mole fraction (mol per mol) between the two experiments. The differences are plotted in terms of the changes in OH attributed to each of the tagging categories, with differences shown between EMIS-01 and 02. §3.2.2 discusses the methane lifetime. Figures 5 and 6 could be improved by plotting the CH4+OH-weighted OH, as discussed in Lawrence et al. (2001, [https://doi.org/10.5194/acp-1-37-2001](https://doi.org/10.5194/acp-1-

37-2001)), to better understand the drivers of methane lifetime changes.

We added a plot showing the global and hemispheric mean tropospheric CH4+OH-weighted OH concentration of individual tagging categories as also suggested by referee 1 to the revised manuscript. We also added a plot showing the zonal mean CH4+OH-weighted total OH of both simulations and their difference (analogously to Figure 5) to the supplement. Figure 6 shows the zonal mean contribution of each tagged OH relative to total OH (in %). We don't think that it adds any value to plot the contribution on the basis on the CH4+OH-weighted total OH instead of the OH mixing ratio as the contribution is given relative to the total OH so that the results would be the same regardless of calculating the contribution on the basis of weighted OH concentration or OH mixing ratios.

3.3 serves as a synthesis section. Breaking this section into subsections—such as "What categories are generally important?" and "How does the importance of these categories change and why?"—would improve focus. Here the discussion becomes somewhat confusing in places, not least because,in the two experiments, some 'categories' involve a change in emissions and consequent change in chemistry, where the authors are easily able to connect the change in emissions to the observed change in ozone column, while in other places the tagging analysis is about processes only, e.g. lightning that has changed between the two experiments, and produced a change in O3. In the latter case, this could be described more fully to explain its impact on O3 columns or methane lifetime.

As mentioned in the comment above, we added a figure showing the global and hemispheric mean tropospheric CH4+OH-weighted OH concentration of individual tagging categories. Using this figure we give some more information on which tagging categories are generally important for the methane loss with OH. For example the figure shows that lightning NOx emissions are the largest individual contributor to tropospheric methane loss with OH. But we want to stress again that online calculated emissions, e.g. lightning NOx emissions, are the same in both simulations, EMIS-01 and EMIS-02 (This is ensured by the use of the QCTM mode, see Methods section). There are differences in lightning NOx emissions between the simulations originally performed for CCMI-1 and CCMI-2022 (see supplement), but these do not influence the simulations analysed in this paper. Therefore, the methane lifetime change attributed to changing lightning NOX emissions as shown in Figure 8 (of the preprint) is small for the category lightning NOx.

We revised Sect. 3.2. and 3.3. to clarify the difference between categories that are generally important for methane loss, and categories that contribute to the methane lifetime difference between the two simulations.

However, in the synthesis in Sect. 3.3. we want to focus on identifying the categories that contribute to the ozone and methane lifetime differences between the two simulations, and on comparing these to the emission changes in the respective categories. We added a sentence to the revised manuscript in Sect. 3.3. to avoid misunderstanding.

The discussion section 4 on limitations and comparison with previous results could be folded into the discussion if the authors choose.

130 We are not quite sure what is suggested here. The discussion consists of the two mentioned sections. Therefore, we prefer to keep the structure as is.

Conclusions are nicely written and add a lot of value.

Thank you very much for this very motivating feedback!

**1 Specific comments**

135

- Figure 1: REFD1 should be capitalized in the caption. Additionally, I could not find a reference to Figure 1 in the text. The simulation names in the legend have been adapted. We reference Figure 1 in line 66, 71 and 73 (of the preprint).

- Figure 3 The labels of the RH colorbars are smaller than neighboring labels, making them difficult to read. The bottom-right panel appears incomplete. Corrected.

140
- Figure 6: The font size on the difference plot colorbar is inconsistent with neighboring labels. Corrected.

- L105 could the authors add an explanation for their choice of labels (rather than keeping the CCMI-1 and CCMI-2022)? The simulations are named differently than CCMI-1 and CMI-2022 because we are analysing simulations that we have performed specifically for this publication, which differ from the simulations that have been originally performed for CCMI-1 and CCMI-2022. The model version that we are using for both, EMIS-01 and EMIS-02, is very similar to the

145 one used for the original simulations performed for CCMI-2022. But this means that, in particular, EMIS-01 does not reproduce the simulations originally performed for CCMI-1 (see Figure 1). We added a short statement in the methods section to explain the naming:

*The simulation set-up is similar to the set-up of the CCMI-2022 REFD1SD simulation (hindcast with specified dynam-ics), but we deviate from the CCMI naming convention to clarify that we have performed the simulations specifically for*
150 *this publication, and that they are not identical to the simulations originally performed for CCMI-1 and CCMI-2022.*

- L107 Clarify whether EMAC uses methane emissions or a boundary condition. If it's the latter, consistent terminology (e.g., "lower boundary") should be used for both $N_2O$ and $CH_4$ in §2.2. EMAC uses a prescribed lower boundary condition for $CH_4$. "Lower boundary" mixing ratios is now used consistently to describe CH4 and N2O.

- L112 Reword 'The influence of the changed prescribed' to 'the influence of the change in the prescribed ODS'? Done.

155
- Table 2 caption needs consistent use of 'non-traffic' Changed to "non-traffic".

- L220 does 'in dependence' mean the dependence of delta tau_CH4,i on delta tau_CH4, OHi? No, here "the dependence of $\tau_{CH_4,OH_{total}}$ on $\tau_{CH_4,OH_i}$", and not on $\Delta\tau_{CH_4,OH_i}$ is meant. The sentence describes the first step of the derivation, which is explained in more detail in Appendix A. We reformulated the paragraph and point now to the corresponding equations in the Appendix to make it easier to understand.

160    – L254-L264 Reword "However, only for land..." to "The TRA and IND..." We rephrased to "However, for land transport and anthropogenic non-traffic emissions ..." as we are not using the acronyms TRA and IND anywhere else in the text.

   – L307 Sentences like "The shipping category also shows..." would benefit from discussing process-level changes (e.g., "In CCMI-1, the tagged OH from X category is larger than in CCMI-2022..."). We made some minor changes to the corresponding paragraph to facilitate readability.

165    – L336 categories occurs twice in the same sentence Removed at the second occurrence.

   – L397 show the increase, rather than show that the increase? We think that the previous formulation is correct as "that the increase ... is steeper ..." is a subordinate clause.

   – L413 EMIS-01 is repeated Corrected.

   – L428 Use "sector" for aviation and biomass burning. We are using sector now for aviation, but we could not find the
170    expression "biomass burning category" in this paragraph.

   – L436 biomass burning sector Done.

   – L440 Replace "present day" with "recent historical." Done.

   – L447 Regional tagging would be a valuable addition, enabling closer examination of regional responses. We have added the sentence as suggested.

**References**

Grewe, V., Tsati, E., Mertens, M., Fromming, C., and Jockel, P.: Contribution of emissions to concentrations: the TAGGING 1.0 submodel based on the Modular Earth Submodel System (MESSy 2.52), Geoscientific Model Development, 10, 2615–2633, https://doi.org/10.5194/gmd-10-2615-2017, 2017.

Jöckel, P., Tost, H., Pozzer, A., Kunze, M., Kirner, O., Brenninkmeijer, C. A. M., Brinkop, S., Cai, D. S., Dyroff, C., Eckstein, J., Frank, F., Garny, H., Gottschaldt, K. D., Graf, P., Grewe, V., Kerkweg, A., Kern, B., Matthes, S., Mertens, M., Meul, S., Neumaier, M., Nutzel, M., Oberländer-Hayn, S., Ruhnke, R., Runde, T., Sander, R., Scharffe, D., and Zahn, A.: Earth System Chemistry integrated Modelling (ESCiMo) with the Modular Earth Submodel System (MESSy) version 2.51, Geoscientific Model Development, 9, 1153–1200, https://doi.org/10.5194/gmd-9-1153-2016, 2016.

Prather, M. J. and Zhu, L.: Resetting tropospheric OH and $CH_4$ lifetime with ultraviolet $H_2O$ absorption, Science, 385, 201–204, https://doi.org/10.1126/science.adn0415, 2024.

Rieger, V. S., Mertens, M., and Grewe, V.: An advanced method of contributing emissions to short-lived chemical species (OH and HO2): the TAGGING 1.1 submodel based on the Modular Earth Submodel System (MESSy 2.53), Geoscientific Model Development, 11, 2049–2066, https://doi.org/10.5194/gmd-11-2049-2018, 2018.